# Pro-Vegetarian Food Patterns and Cancer Risk among Italians from the Moli-Sani Study Cohort

**DOI:** 10.3390/nu15183976

**Published:** 2023-09-14

**Authors:** Claudia Francisca Martínez, Augusto Di Castelnuovo, Simona Costanzo, Teresa Panzera, Simona Esposito, Chiara Cerletti, Maria Benedetta Donati, Giovanni de Gaetano, Licia Iacoviello, Marialaura Bonaccio

**Affiliations:** 1Department of Epidemiology and Prevention, IRCCS NEUROMED, 86077 Pozzilli, Italy; claudia.martinez@moli-sani.org (C.F.M.); simona.costanzo@moli-sani.org (S.C.); teresa.panzera@moli-sani.org (T.P.); simona.esposito@moli-sani.org (S.E.); chiara.cerletti@moli-sani.org (C.C.); mbdonati@moli-sani.org (M.B.D.); giovanni.degaetano@moli-sani.org (G.d.G.); marialaura.bonaccio@moli-sani.org (M.B.); 2Population Health Research Center, National Institute of Public Health, Cuernavaca 62100, Mexico; 3Mediterranea Cardiocentro, 80122 Naples, Italy; dicastel@ngi.it; 4Department of Medicine and Surgery, Research Center in Epidemiology and Preventive Medicine (EPIMED), University of Insubria, 21100 Varese, Italy

**Keywords:** pro-vegetarian food patterns, plant-based diets, cancer risk, cancer hospitalization

## Abstract

Besides the Mediterranean diet, there is a paucity of studies examining plant-based diets in relation to cancer outcomes in Mediterranean populations. We analyzed 22,081 apparently cancer-free participants (mean age 55 ± 12 year) from the Moli-sani study (enrollment period 2005–2010; Italy). A general pro-vegetarian food pattern was computed by assigning positive or negative scores to plant- or animal-derived foods, respectively from a 188-item FFQ. *A priori* healthful or unhealthful pro-vegetarian food patterns distinguished between healthy plant foods (e.g., fruits, vegetables) and less-healthy plant foods (e.g., fruit juices, refined grains). Cancer incidence was defined as the earliest diagnosis of cancer from hospital discharge records over a median follow-up of 12.9 years. In multivariable-adjusted analyses, a general pro-vegetarian food pattern was associated with a lower rate of cancer incidence (HR = 0.85; 95%CI 0.75–0.97 for Q5 vs. Q1); no association was observed between the healthful or unhealthful pro-vegetarian food patterns and overall cancer incidence. A healthful pro-vegetarian pattern, however, was inversely associated with digestive cancer (HR = 0.76; 95%CI 0.58–0.99 for Q5 vs. Q1), while the unhealthful pro-vegetarian pattern was directly linked to respiratory cancer (HR = 1.68; 95%CI 1.06–2.68 for Q5 vs. Q1). Our findings in a Mediterranean population support the hypothesis that some, but not all pro-vegetarian diets, might prevent some cancers.

## 1. Introduction

Cancer represents a leading cause of premature death in Europe and is progressively surpassing cardiovascular diseases (CVD) as the principal cause of death [1]. However, it has been estimated that 30–50% of all cancer cases are preventable both by modifying or avoiding key risk factors (e.g., smoking, alcohol abuse) and through a healthy diet and lifestyle [2]. Regarding diet, the World Cancer Research Fund/American Institute for Cancer Research strongly recommends, as a key strategy for cancer prevention, adherence to a balanced nutritious diet rich in plant-derived foods (e.g., whole grain, vegetables, fruit, and beans), while limiting intakes of red and processed meat, fast foods, and other processed foods high in fats, starches, or sugars [3].

Several healthful dietary patterns are favorably associated with cancer outcomes. These are mostly plant-based diets characterized by a large amount of food (e.g., fruits, vegetables, nuts, and seeds), that are natural sources of bioactive compounds with potential anti-cancer activities [4].

To date, a number of studies have examined the potential advantages of a traditional Mediterranean diet against certain cancers, but evidence varies greatly according to different cancer sites, with moderate evidence being reached only for cancer mortality and colorectal cancer risk, while being rated as “low” or “very low” for other cancer subtypes (e.g., bladder, breast, gastric, prostate) [5]. Although mainly based on the consumption of plant-derived foods, the Mediterranean diet still includes moderate consumption of animal-derived foods, such as poultry, fish, and dairy products (mostly in the form of long-preservable cheese), as well as moderate amounts of wine with meals [6].

On the contrary, strictly vegetarian diets completely exclude meat, meat-derived foods, and other products obtained from animals [7]. They are reported to be favorably associated with several health outcomes [8], but their impact on cancer is still controversial.

Although the proportion of true vegetarians is relatively scarce in most populations [9], the concept of pro-vegetarian diets, emphasizing the consumption of plant-derived foods, but allowing limited amounts of animal-based products [10], has gained increasing attention over the last years.

To estimate adherence to a pro-vegetarian food pattern distinct from full vegetarianism, a general pro-vegetarian food pattern index was first developed, weighing both plant-derived and animal-derived foods [9]. Subsequently, two additional indices were developed to examine the quality of the plant-based foods by distinguishing *a priori* between *healthful* and *unhealthful* pro-vegetarian food patterns, based on existing literature data [11]. In fact, not all plant foods might be equally healthy due to their different nutritional composition and the extent of food processing. This led to the differentiation of the quality of plant-based diets, as a healthful plant-based diet prioritizes the consumption of, e.g., vegetables, fruit, and whole grains, and an unhealthful plant-based diet mainly comprises a large amount of fruit drinks and juices, potatoes (mostly in the form of French fries and chips), sugary beverages, and cereals made with refined grains [11].

In the last 10 years, several large cohort studies supported the health advantages of a plant-based/pro-vegetarian diet, mostly in relation to mortality [9,12,13,14,15,16] and to the incidence of major chronic diseases, including CVD [13,17] and type 2 diabetes [11]. Research examining the association of *a priori*-defined plant-based indices with risk of cancer (all or site specific) [18,19,20,21,22] or cancer death [12] is less robust and has yielded inconsistent findings. Also, most of the available evidence stems from US populations, with limited research in Mediterranean populations. Moreover, not all cohort studies distinguished the quality of plant foods in relation to cancer outcomes.

To fill this knowledge gap and contribute to clarifying the uncertain relationship between plant-based diets and cancer risk, we sought to examine the association of three previously defined pro-vegetarian food patterns, defined as general, healthful, and unhealthful, in relation to cancer incidence in a general population residing in a Southern Italian region.

## 2. Methods

### 2.1. Study Population

We analyzed data from the Moli-sani study, which is a prospective population-based cohort consisting of 24,325 men and women aged ≥35 years, who were randomly recruited from the general population of the Molise region in Southern Italy. The cohort was established in 2005–2010 with the main aim of investigating the contribution of environmental and genetic risk factors in the onset of major non-communicable diseases and mortality. Pregnancy at the time of recruitment, disturbances in mental or decision-making impairments, current poly-traumas or coma, or refusal to sign the informed consent were exclusion criteria. The study design was described previously [23]. From these analyses, we excluded those participants with a prevalent registered or self-reported malignant cancer, had missing data on diet, reported extreme energy intakes (<800 or >4000 kcal/d in men and <500 or >3500 kcal/d in women), their dietary or medical questionnaires were judged as unreliable by interviewers or were lost to follow-up. After exclusions, a total of 22,081 cancer-free participants were included in the analyses (Appendix A). The Moli-sani study complies with the Declaration of Helsinki and was granted the approval of the Ethics Committee of the Catholic University in Rome, Italy, ID Prot. pdc. P.99 (A.931/03-138-04)/C.E./2004. Written informed consent was collected from all participants.

### 2.2. Dietary Data Collection and Computation of Three Different Pro-Vegetarian Food Patterns

At baseline visit, data on participants’ dietary intake in the preceding year were assessed by an interviewer-administered, semi-quantitative EPIC (European Prospective Investigation into Cancer and Nutrition) food frequency questionnaire (FFQ), which was validated and adapted to the Italian population [24]. The FFQ contained 14 sections (i.e., pasta/rice, soup, meat, fish, raw vegetables, cooked vegetables, eggs, sandwiches, salami, and other cured meats, cheese, fruit, bread/wine, milk/coffee/cakes, and herbs/spices) with 248 questions regarding 188 different foods [24]. Using specialized software, the frequency and quantity of each food item were linked to Italian Food Tables [25,26] to estimate daily energy intake and macro- and micro-nutrients.

We computed a general pro-vegetarian food pattern which assigned positive scores to plant-derived foods and reverse scores to animal-derived foods, as proposed by Martinez-Gonzalez et al. [9]. For the healthful and unhealthful pro-vegetarian food patterns, we used as reference the work by Satija et al. [11], and the modification proposed by Oncina et al., which evaluated only olive oil instead of all vegetable oils [21]. The three patterns were determined by using dietary information of 17 food groups (5 animal-derived and 12 plant-derived food groups), which are all described in Table 1. To compute the three pro-vegetarian food patterns, the consumption (g/d) of food groups was adjusted for total energy for men and women, respectively using the residual method. The energy-adjusted residuals were grouped according to quintiles. To obtain the general pro-vegetarian food pattern, quintile values of 7 plant-derived foods (i.e., fruits, legumes, whole grain products, refined grain products, potatoes, nuts and dried fruit, olive oil) and reverse quintile values of 5 animal-derived foods (i.e., meat and meat products, animal fats, eggs, fish and seafood, milk and dairy) were summed. The final score of the general pro-vegetarian food pattern potentially ranged from 12 (minimum adherence) to 60 (maximum adherence). To compute the healthful pro-vegetarian food pattern, positive scores were given to 7 healthy plant food groups (i.e., fruits, vegetables, legumes, whole grains, nuts and dried fruit, olive oil, tea and coffee), and reverse scores to 5 less-healthy plant food groups (i.e., refined grains, potatoes, fruit juices, sugar-sweetened beverages, sweets and desserts). For the unhealthful pro-vegetarian food pattern, less-healthy plant-derived foods were assigned positive scores, whereas reverse scores were given to the healthy plant-derived food groups and foods of animal origin.

The five animal-derived food groups (i.e., meat and meat products, animal fats, eggs, fish and seafood, and milk and dairy products) were scored negatively both in healthful and unhealthful pro-vegetarian food patterns, whose final scores potentially ranged from 17 to 85 points (Table 1).

Adherence to a traditional MD was measured through the *a priori* 9-item Mediterranean diet score (MDS) proposed by Trichopoulou et al. [27], which reflects the dietary habits of the olive tree growing throughout countries bordering the Mediterranean Sea basin.

### 2.3. Baseline Covariate Assessment

At study entry, participants completed an interview that included sociodemographic information, lifestyle, and clinical variables. Based on the highest qualification attained, education was categorized as (1) up to lower secondary school (approximately ≤8 years), (2) upper secondary school (>8–13 years), and (3) post-secondary education (>13 years). Personal history of cardiovascular disease (angina, myocardial infarction, revascularization procedures, peripheral artery diseases, and cerebrovascular events) was self-reported and confirmed by clinical records and therapy. Comorbidities, such as hypertension, hyperlipidemia, or diabetes, were defined if the participant reported having a treatment with disease-specific drugs. Regular use of low-dose aspirin for primary or secondary CVD prevention (no/yes) was recorded. Leisure-time physical activity for sport, walking, and gardening was assessed by a structured interviewer-administered questionnaire [28] and expressed as daily energy expenditure in metabolic equivalent task hours (MET-h/d) [29]. Height and weight were measured to estimate body mass index (BMI) that was classified as normal (≤25 kg/m^2^), overweight (>25 and <30 kg/m^2^), or obese (≥30 kg/m^2^). Participants’ smoking habit was categorized as never, current, or former smoker (reporting to have not smoked in the previous 12 months or more at the time of interview). Housing tenure was classified as rented, ownership of one dwelling, and ownership of more than one dwelling. Marital status was coded as married/live-in couple, separated/divorced, single, and widower. Additional covariates were considered for females only and included the menopausal status, use of oral contraceptives, and use of hormone replacement therapy.

### 2.4. Outcome Definition and Assessment

The outcome examined in this study was cancer incidence, which was defined as the earliest diagnosis of cancer from hospital discharge records. A hospitalization was defined as any length of stay of at least 24 h in a hospital or clinic. If a patient was transferred to another hospital or facility, it was considered a single hospitalization. Hospitalizations for rehabilitation and chemotherapy and/or radiotherapy (i.e., elective treatment in a hospital-based unit) were excluded. Primary and secondary diagnoses for hospitalization were coded using the 9th version of the International Classification of Diseases. A cancer hospitalization was assessed if the primary or the first secondary diagnoses of admission to the hospital were coded as ICD-9: 140–172, 174–208. The Moli-sani study cohort was followed-up for cancer hospitalization from study entry (2005–2010) through 31 December 2020.

## 3. Statistical Analysis

We quantified differences in the distribution of baseline covariates across fifths of a general pro-vegetarian food pattern using generalized linear models for both continuous and categorical variables adjusted for age, sex, and energy intake (kcal/d), (GENMOD procedure for categorical variables and GLM procedure for continuous variables using SAS software). Associations of three pro-vegetarian food patterns (both as continuous or categorical dependent variables) with the rate of cancer hospitalization were calculated as hazard ratios (HRs) and 95% confidence intervals (CIs) using Cox proportional hazard models. No violations of the proportional hazard assumption (log(−log) survival plots curves) were visually identified.

Potential confounders (measured at baseline) were defined *a priori* and identified based on existing literature, rather than deferring to statistical criteria [30], on the basis of their previously documented association both with diet and cancer risk [31,32,33]. The following multivariable models were ultimately fitted: Model 1 included age at baseline, sex, energy intake (kcal/d), and alcohol intake (g/d; continuous). Model 2, as in Model 1, further controlled for educational level, housing tenure, place of residence, smoking, body mass index (categorical), leisure-time physical activity, history of CVD, diabetes, hypertension, hyperlipidemia (no/yes), aspirin use, hormone replacement therapy, oral contraception, menopausal status, family history of cancer. Participants contributed person-time until the date of cancer hospitalization, death, date of emigration or loss to follow-up, or end of follow-up, whichever occurred first. We used cancer-free survival curves (adjusted for potential confounding variables as in Model 2) to describe cancer incidence over time across categories of adherence to pro-vegetarian food patterns. To test the robustness of the results, several sensitivity analyses were conducted by excluding (a) participants with a history of CVD, (b) participants with diabetes, and (c) cancer hospitalizations that occurred within the first 2 years of follow-up. Main analyses (Model 2) were re-run in men and women, and across age groups (<65 and ≥65 years), BMI categories, and by smoking status. Appropriate terms for testing interaction were included in the multivariable model to test for a difference of effect of the pro-vegetarian food patterns (1 SD increase) in relation to cancer risk within pre-specified population groups.

Missing data on covariates were handled using multiple imputation (SAS PROC MI, followed by PROCMIANALYZE), over different simulations (n = 10 imputed datasets) to avoid bias introduced by not-at-random missing data patterns. The data analysis was generated using SAS/STAT software, version 9.4 (SAS Institute Inc., Cary, NC, USA).

## 4. Results

Sociodemographic characteristics of the analyzed sample across fifths of adherence to a general pro-vegetarian food pattern are shown in Table 2. The median value of the general pro-vegetarian food pattern in this population was 36 (mean = 36; SD = 5.6; min-to-max = 14 to 57), median values for the healthful and unhealthful pro-vegetarian food patterns were 47 (mean = 47; SD = 6.0; min-to-max = 28 to 69) and 47 (mean = 47; SD = 6.5; min-to-max = 22 to 70), respectively. The general pro-vegetarian was directly associated with the healthful pro-vegetarian food pattern (Spearman correlation coefficient = 0.72; *p*-value <0.0001) and showed a weak inverse relationship with an unhealthful pro-vegetarian food pattern (Spearman correlation coefficient = −0.03; *p*-value <0.0001); the healthful and unhealthful pro-vegetarian food pattern were moderately correlated (Spearman correlation coefficient = −0.40; *p*-value <0.0001). The Spearman correlation coefficients of a general, healthful, and unhealthful pro-vegetarian food pattern with the Mediterranean diet score were 0.52, 0.49, and −0.30, respectively (all *p*-values <0.0001), suggesting weak to moderate strengths of these associations [34].

Subjects with higher adherence to the general pro-vegetarian food pattern tended to have higher adherence to the traditional Mediterranean diet, to be older, to have higher household income, and were more physically active. However, participants following a general pro-vegetarian food pattern were more likely to report a history of cardiovascular disease, hypertension, hyperlipidemia, and daily aspirin use than those with lower adherence to this food pattern (Table 2).

In terms of consumption, the general pro-vegetarian was directly associated with intake of plant-based food, and inversely with animal-based food consumption (Appendix A). Increasing adherence to the healthful pro-vegetarian food pattern was positively associated with total energy intake, dietary fiber, alcohol (g/d), and carbohydrate and fat (% of total energy intake; TEI); higher adherence to an unhealthful pro-vegetarian food pattern was linked to reduced total energy intake, dietary fiber, and fat intake (%TEI), and higher daily energy from carbohydrate and alcohol intake (g/d) (Appendix A). Over 12.9 years (median) of follow-up (interquartile ranges = 11.8 to 14.1 years; 271,891 person-years), we documented a total of 2306 hospital admissions for cancer (any site). Of these, 598 were digestive cancers (25.9% of all cancer hospitalizations; ICD-9 150–159); 183 (7.9%) respiratory and intrathoracic organs (ICD-9 160–165); 395 (17.1%) genitourinary organs (ICD-9 179–189); 285 (12.4%) breast cancer (ICD-9 174); 219 (9.5%) prostate (ICD-9 185); 178 (7.7%) lymphatic and hematopoietic tissue (ICD-9 200–209); 41 (1.8%) brain and nervous system (ICD-9 191–192); 407 (17.6%) were other cancer types. In multivariable-adjusted analyses controlled for known risk factors, higher adherence to the general pro-vegetarian food pattern was associated with 15% lower rate of cancer hospitalization (HR = 0.85; 95%CI 0.75–0.97 for Q5 (score > 42) vs. Q1 (score < 31); *p*-value for trend across fifths = 0.045). Similarly, an increment of 1 SD in the general pro-vegetarian food pattern was linked to a reduction of 5% in cancer rate (HR = 0.95; 95%CI 0.91–0.99) (Table 3). Further adjustment for the MDS did not alter these associations (HR = 0.84; 95%CI 0.72–0.97 for Q5 vs. Q1 of the general pro-vegetarian food pattern; HR = 0.94; 95%CI 0.90–0.99 for 1 SD increment in the pro-vegetarian food pattern). Multivariable-adjusted cancer-free survival curves across categories of the general pro-vegetarian food pattern were well separated with a tendency to diverge over time (Figure 1). Neither healthful nor unhealthful pro-vegetarian food patterns were associated with overall cancer risk (Table 3; Appendix A).

### Subgroup and Sensitivity Analyses

None of the baseline risk factors act as an effect modifier of the association between the pro-vegetarian food patterns (general, healthful, or unhealthful) with cancer risk (Table 4). Sensitivity analyses showed consistent inverse associations between a general pro-vegetarian food pattern with cancer risk in all the analyzed scenarios (excluding baseline CVD, diabetes, or early cancer hospitalizations) (Table 4). When hospitalizations were separately examined by specific cancer site, a general pro-vegetarian food pattern was associated with a lower rate of respiratory cancer (HR = 0.61; 95%CI 0.38–0.98 for Q5 vs. Q1), or digestive neoplasm (HR = 0.74; 95%CI 0.57–0.95 for Q5 vs. Q1) (Table 5). For the healthful pro-vegetarian food pattern, a lower rate of digestive cancer was also observed (HR = 0.76; 95%CI 0.58–0.99 for Q5 vs. Q1). Higher adherence to an unhealthful pro-vegetarian food pattern was otherwise directly associated with the rate of respiratory tract cancer (HR = 1.68; 95%CI 1.06–2.68 for Q5 vs. Q1) (Table 5).

## 5. Discussion

In this large prospective cohort of Italian men and women, who were apparently without cancer at baseline, a higher adherence to the general pro-vegetarian food pattern was associated with a lower rate of cancer incidence overall, specifically for digestive or respiratory cancers. Adherence to healthful or unhealthful pro-vegetarian food patterns was not associated with overall cancer risk. However, when specific cancer sites were considered, a healthful pro-vegetarian food pattern was linked to a reduced rate of digestive cancer, whereas an unhealthful pro-vegetarian food pattern was associated with a higher likelihood of respiratory cancers.

The relationships between plant-derived foods and diets with cancer risk have currently different levels of evidence, based on the food group and the cancer site. Probably, some key food groups that are positively scored in the general and healthful pro-vegetarian food patterns (e.g., dietary fiber and whole grains) could reduce the risk of colorectal cancer, while coffee consumption (that is positively scored in the healthful pro-vegetarian diet) has been associated with lower incidence of liver cancer [3]. Limited evidence exists on the potential protective effects of diets rich in fruits, vegetables, polyphenols, and vitamin C against respiratory cancer [3], while foods with high-glycemic index, such as sugary beverages, cereals made with refined grains, fruit juices and potatoes, that scored high in the unhealthful pro-vegetarian diet, could possibly increase the risk of lung cancer [35], although evidence is not conclusive [36].

Also, we acknowledge that although the total number of cancer hospitalizations was considerably large, the number of specific cancers was restricted; therefore, possibly limiting the statistical power of analyses for different cancer sites (e.g., brain cancer).

Of interest, in our study, the inverse association of a general pro-vegetarian food pattern with overall cancer risk was independent of adherence to the Mediterranean diet. This is possibly due to the fact that there are some important differences between the pro-vegetarian food pattern and its healthful version and the traditional Mediterranean diet, mostly in relation to the scoring criteria for three foods: potatoes, fish, and alcohol. This is further supported by the moderate correlation observed between pro-vegetarian and Mediterranean diet scores, indicating that these novel dietary indices might be able to capture novel unique relations of a pro-vegetarian diet with respect to cancer outcomes [37].

This is possibly the first study that has been conducted on Italians to examine three different pro-vegetarian food patterns in relation to cancer risk and acknowledge the quality of plant-derived foods.

Our findings on the general pro-vegetarian food pattern and cancer risk are in line with prior analyses in the large NutriNet-Santè cohort in France, showing a reduced risk of overall cancer for participants with strict adherence to a pro-vegetarian diet [18]. As shown in our study and in this large French cohort, a general pro-vegetarian food pattern was mostly effective in reducing the risk of digestive or respiratory cancers. This is possibly due to the high intake of fruits, vegetables, and fiber reported by participants with higher adherence to the overall pro-vegetarian food pattern, with a potential protective role against carcinogenesis [34,35]. In the UK Biobank study, a difference in overall cancer risk was observed between healthy and unhealthy plant-based diets, but not for individual cancers [21]. However, the potential effectiveness of a healthful plant-based diet against digestive cancers was documented in a previous hospital-based, multi-case control study [21], that showed a substantial reduction in the risk of oesophagus, stomach, and pancreatic cancers for participants with a very high adherence to this food pattern. Consistently, a large dietary share of unhealthful plant-based products was associated with a higher likelihood of oesophagus and stomach cancers, but not pancreatic neoplasm [21]. In our analyses, we did not observe any association of these three pro-vegetarian food patterns with breast or prostate cancers. In this study, our data are in contrast with analyses on a large sample of Spanish women from the SUN cohort, where a moderate adherence to a pro-vegetarian diet was associated with lower breast cancer incidence compared to lower consumption, and mostly amongst pre-menopausal women. In addition, a moderate conformity to an unhealthful pro-vegetarian food pattern was associated with a two-fold increased breast cancer risk, but only amongst post-menopausal women [19]. However, the lack of an association between a pro-vegetarian diet with breast cancer was reported amongst women participants in the NutriNet-Santè cohort from France [18].

Prospective analyses from the Health Professionals Follow-Up Study in US supported evidence that greater consumption of healthful plant-based foods was associated with a lower risk of prostate cancer, specifically aggressive forms among men aged <65 years [20]. However, analyses from the NutriNet-Santè cohort did not highlight a role of a general pro-vegetarian against prostatic neoplasm [18].

A relatively low number of cohort studies have analyzed plant-based diets in relation to cancer death, but the evidence is not conclusive. A prospective cohort study conducted in the US found an inverse association of an overall plant-based diet with cancer mortality, but the quality of plant foods was unrelated to the outcome [15]. Similarly, there were no meaningful associations between changes in plant-based diet indices and cancer mortality in a well-established large cohort of US men and women [12]. Finally, analyses on a large cohort of South Korean adults showed a higher risk of cancer death for participants with the highest adherence to a diet rich in unhealthful plant-derived foods, while overall or healthful plant-based diets seemed unrelated to the outcome [13].

There are several potential mechanisms through which greater adherence to a plant-based diet may influence cancer risk. High amounts of vegetables, fruits, and legumes increase the dietary intake of phytochemicals and vitamins with antioxidant and antiproliferative activity [12,38,39,40]. In addition, higher consumption of fiber and whole grains [41,42,43], and low consumption of sugar-sweetened beverages may play an important role in regulating mechanisms associated with glucose control and insulin growth factor, which are related to a higher risk of cancer [44,45,46,47,48]. In addition, fiber from a plant-based diet constitutes a beneficial substrate for intestinal microbiota and metabolites associated with certain cancers [49]. Also, both the general and healthful pro-vegetarian food patterns emphasize the consumption of olive oil, which has long been associated with lowering the risk of developing or dying from cancer, possibly through its anti-inflammatory, antioxidant, anti-atherogenic, and anti-thrombotic properties [50].

The health advantages of a pro-vegetarian diet could also be due to a low dietary share of animal-based foods; indeed, elevated intakes of some animal-derived foods, such as red and processed meat, have been reportedly associated with overall cancer risk [51,52] and certain cancers [52,53]. On the other hand, sugary beverages have been associated with increased incidence of and mortality from breast cancer among post-menopausal women [44,46,54]. Also, evidence suggests an increased risk of colon and gastric cancer associated with refined grains; however, findings remain unclear [52].

Other detrimental effects, aside from the poor quality of some unhealthful plant-based foods, could be due to non-nutritional factors, such as increased exposure to additives, toxic contaminants migrated from food packaging, and exposure to acrylamide, which is produced during cooking procedures at high-temperature and may increase cancer risk [55,56,57]. In fact, the unhealthful plant-based diets are also characterized by the presence of ultra-processed foods as defined by the Nova classification [58], and thus share all the critical features of these foods which go well beyond their poor nutritional composition [59]. Therefore, analyses (and interpretations) on the quality of plant-based diets in relation to health outcomes, including cancer, should not be limited to the evaluation of the nutritional content of plant-derived foods, but should consider the extent of food processing, as well.

## 6. Strengths and Limitations

The strengths of this study include the large sample size, a prospective population-based cohort design, a follow-up of more than 10 years, and the large number of covariates used to minimize, at least in part, confounding. Moreover, we computed three indices of pro-vegetarian food patterns that are easily reproducible for further comparisons. Also, we used sensitivity analyses to support the robustness of the results, by excluding participants who might have modified habitual dietary intakes due to illness.

Several limitations should be considered when interpreting our findings. The possibility of residual confounding remains, given the observational design, so that any causality relationship is only proposed. However, the main potential confounders were considered; therefore, it is unlikely that residual confounding entirely explains the observed results. Limitations in the outcome definition should be also acknowledged, since in this study, we only analyzed cancer hospitalizations, and consequently, cases of cancer that were managed in outpatient settings were missed.

Additionally, interpretation of risk estimations for specific cancer sites should consider the restricted number of events in subgroups, which may affect the statistical power and precision of CIs. Another weakness is the lack of repeated dietary assessments during follow-up, although dietary habits in adulthood tend to remain stable over time [60]. Moreover, dietary data were self-reported with an inherent risk of error and bias, possibly attenuated by exclusion of participants with implausible energy intakes. Furthermore, the FFQ used in the present study was extensively validated against diet records and biomarkers [61]. We also acknowledge that the association of each dietary component with cancer risk was not examined since diet was considered as a synergistic interaction of multiple components. Although based on prior evidence, the identification of healthful plant foods has an element of subjectivity [11]. Moreover, we could not differentiate potatoes by cooking method or processing level; therefore, all types of potatoes were scored as “unhealthy” in the healthful pro-vegetarian food pattern. The lack of correction for family-wise error is another limitation of this study. Finally, the generalizability of the results could be limited to an Italian population.

## 7. Conclusions

Our findings in a Mediterranean population suggest that, as already known for a traditional Mediterranean diet, a pro-vegetarian diet, based on healthful plant-derived foods and limited animal-derived products intakes, could be effective in reducing cancer incidence. Moreover, our data support prior evidence showing that a large dietary share of unhealthful, though plant-based foods, mostly highly processed, is associated with an increased risk of certain cancers [19,22] or cancer death [13,15,62].

In conclusion, a diet that contains high amounts of healthful plant products and minimal amounts of animal products should be encouraged, when a traditional Mediterranean diet cannot be followed. Future targeted analyses could be helpful to understand whether the observed health advantages associated with this pro-vegetarian dietary pattern are mostly due to the large amount of plant foods or to the lower consumption of animal foods or both. In addition, studies could attempt to establish the ideal ratios of high plant food consumption and low animal food consumption to maximize health benefits. In all instances, it is important in public health recommendations and dietary guidelines worldwide to discourage the consumption of unhealthful, especially ultra-processed, plant-derived products.

## Figures and Tables

**Figure 1 nutrients-15-03976-f001:**
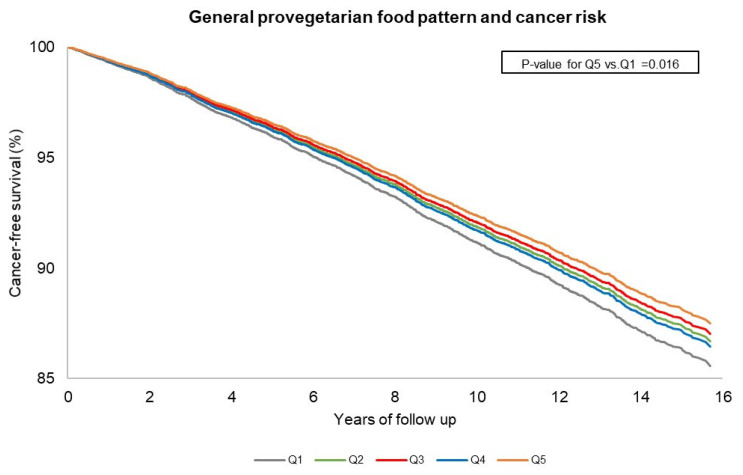
Association between adherence to a general pro-vegetarian food pattern with cancer risk in participants from the Moli-sani study cohort (n = 22,081). Legend for Figure 1. Cancer-free survival curves were obtained from the multivariable model adjusted for age, sex, energy intake, alcohol intake, residence, educational level, housing tenure, occupational class, smoking status, body mass index, leisure-time physical activity, history of CVD, diabetes, hypertension, hyperlipidemia, aspirin use, oral contraception use, hormone replacement therapy, menopausal status, and family history of cancer. Cancer-free survival curves were generated using the first imputed dataset. The other imputed datasets were similar, and thus omitted.

**Table 1 nutrients-15-03976-t001:** Scoring criteria for the pro-vegetarian food patterns in the Moli-sani study cohort, Italy, 2005–2010.

Component	Included Foods	General Pro-Vegetarian	Healthful Pro-Vegetarian	Unhealthful Pro-Vegetarian
*Plant food groups (n = 12)*				
1. Vegetables	Spinach, turnip greens, salad, green pepper, pumpkin, tomatoes, carrot, beet, broccoli, brussel sprouts, cauliflower, cabbage, kale, mushrooms, garlic, onions, zucchini, artichoke, fennel, olives	Positive	Positive	Reverse
2. Fruits	Citrus, apple, pear, banana, kiwi, grape, peach, apricot, prune, strawberries, melon, fruit salad, figs, cherries, persimmon	Positive	Positive	Reverse
3. Legumes	Beans, chickpeas, lentils, peas, broad beans	Positive	Positive	Reverse
4. Whole grain	Whole grain bread	Positive	Positive	Reverse
5. Refined grains	Crispbread/rusks, breakfast cereals, white bread, other bread, rice, pasta, and other grains	Positive	Reverse	Positive
6. Potatoes	Potatoes	Positive	Reverse	Positive
7. Nuts and dried fruit	Walnut, hazelnut, almond, peanut, dried fruit	Positive	Positive	Reverse
8. Olive oil	Common olive oil	Positive	Positive	Reverse
9. Tea and coffee	Tea, caffeinated coffee, decaffeinated coffee	Not scored	Positive	Reverse
Fruit juices	Fruit juices	Not scored	Reverse	Positive
Sugar-sweetened beverages	Carbonated/soft/isotonic drinks, diluted syrups	Not scored	Reverse	Positive
Sweets and desserts	Chocolate, nut spread, candies, cakes, pies, pastries, puddings (non-milk based), biscuits, dry cakes, honey, jam, and sugar	Not scored	Reverse	Positive
*Animal food groups (n = 5)*				
Meat and meat products	Chicken or turkey, rabbit, pork, beef, lamb, veal, offal, ham, cured meats, salami, mortadella, sausage, hamburger	Reverse	Reverse	Reverse
Animal fats for cooking or as a spread	Butter, another animal fat	Reverse	Reverse	Reverse
Eggs	Eggs	Reverse	Reverse	Reverse
Fish and other seafood	Hake, sole, sardines, trout, swordfish, shrimp, prawns, squid, cuttlefish, octopus, clams, stock fish, canned fish	Reverse	Reverse	Reverse
Milk and dairy products	Whole milk, partially-skimmed or skimmed milk, plain yogurt, low-fat yogurt, fruit yogurt, hard cheese, soft cheese, ice cream	Reverse	Reverse	Reverse

Positive scores indicate that a higher consumption of this food group receives higher scores. Reverse scores indicate that a higher consumption of this food group receives lower scores. In the general pro-vegetarian food pattern, consumption of whole grains and refined grains was aggregated as the “grains” food group.

**Table 2 nutrients-15-03976-t002:** Baseline characteristics of study participants across levels of adherence to a general pro-vegetarian food pattern (FP) in the Moli-sani study cohort (n = 22,081) ^1^.

		General Pro-Vegetarian Food Pattern (Quintiles of)					
	All	Q1	Q2	Q3	Q4	Q5	
General pro-vegetarian FP (median, IQR)	36 (32–40)	29 (27–30)	33 (32–34)	36 (35–37)	39 (38–40)	43 (42–45)	<0.0001
Healthful pro-vegetarian FP (median, IQR)	47 (43–51)	41 (38–44)	44 (42–47)	47 (44–50)	49 (46–52)	53 (50–56)	<0.0001
Unhealthful pro-vegetarian FP (median, IQR)	47 (43–51)	47 (43–52)	47.5 (43–52)	47 (42–51)	47 (42–51)	47 (43–51)	<0.0001
Mediterranean diet score (median, IQR)	4.0 (1.6)	3.0 (2.0–4.0)	4.0 (3.0–5.0)	4.0 (3.0–5.0)	5.0 (4.0–6.0)	6.0 (5.0–7.0)	<0.0001
No. of subjects (%)	22,081	4720 (21.4)	4042 (18.3)	4670 (21.2)	4037 (18.3)	4612 (20.9)	-
Age (years; mean, SD)	55.2 (11.7)	52.1 (11.2)	54.6 (11.8)	55.6 (11.7)	56.5 (11.6)	57.2 (11.3)	<0.0001
Men	48.0	48.5	46.8	48.3	47.7	48.5	0.47
Urban residence	67.0	66.1	66.6	66.8	67.4	68.2	0.64
Educational level							0.073
Up to lower secondary	52.1	50.1	51.9	51.7	53.4	53.6	
Upper secondary	35.0	36.6	35.3	35.3	34.2	33.2	
Post-secondary	12.9	13.2	12.7	12.9	12.2	13.1	
Missing data	0.1	0.02	0.1	0.1	0.2	0.1	
Housing							<0.0001
Rent	8.9	10.1	9.3	8.7	8.3	8.0	
One dwelling ownership	82.2	83.0	83.2	82.1	82.5	80.2	
More than one dwelling ownership	8.9	6.7	7.4	8.9	9.1	11.6	
Missing data	0.2	0.2	0.1	0.3	0.1	0.2	
Occupational class							0.0032
Professional/managerial	20.7	19.9	19.6	20.7	21.1	21.9	
Skilled non-manual occupations	36.5	37.6	37.0	37.1	36.3	34.3	
Skilled manual occupations	18.0	19.5	17.8	17.2	17.4	17.9	
Partly skilled/Unskilled	18.7	17.0	19.5	19.0	19.3	19.1	
Unemployed/unclassified	6.1	6.0	6.1	6.0	5.9	6.8	
Smoking status							0.10
Non-smokers	49.5	47.5	50.8	48.1	50.5	51.2	
Current	23.3	26.8	24.0	23.8	21.1	20.4	
Former	27.1	25.7	25.1	28.0	28.3	28.3	
Missing data	0.1	0.0	0.1	0.1	0.1	0.1	
Leisure-time PA, MET-h/day (mean, SD) ^2^	3.6 (4.0)	3.3 (3.9)	3.3 (3.7)	3.5 (3.9)	3.7 (4.1)	4.0 (4.5)	<0.0001
Body mass index							0.12
Normal weight	27.7	30.2	28.1	26.1	26.8	27.1	
Overweight	42.9	41.4	43.2	43.4	43.2	43.3	
Obese	29.3	28.3	28.6	30.4	29.9	29.5	
Missing data	0.1	0.1	0.1	0.1	0.1	0.1	
Cardiovascular disease							0.0043
Yes	5.0	3.4	4.1	5.5	5.6	6.6	
Missing data	1.6	1.4	1.3	1.4	1.9	1.8	
Diabetes							0.0001
Yes	4.7	3.9	5.0	5.4	5.1	4.1	
Missing data	1.3	1.4	1.3	1.1	1.3	1.1	
Hypertension							0.027
Yes	27.8	21.6	25.6	29.1	31.2	31.9	
Missing data	0.7	0.8	0.8	0.8	0.6	0.6	
Hyperlipidemia							<0.0001
Yes	7.6	4.7	6.5	8.0	8.6	10.3	
Missing data	0.9	0.7	0.8	0.8	0.8	1.3	
Aspirin use							0.030
Yes	4.6	2.8	4.1	4.7	5.7	5.7	
Missing data	1.9	1.6	1.6	1.8	1.8	2.5	
Menopausal status							0.91
Yes	56.4	46.4	54.4	57.3	60.9	64.1	
Missing data	0.1	0.0	0.1	0.2	0.1	0.1	
Hormone replacement therapy							0.028
Yes	5.6	4.7	4.6	6.6	5.6	6.4	
Missing data	0.01	0.0	0.05	0.0	0.0	0.0	
Oral contraception use							0.72
Yes	28.2	31.8	28.2	28.8	26.4	25.6	
Missing data	0.02	0.04	0.05	0.0	0.0	0.0	
Family history of cancer	40.4	39.1	40.2	40.6	40.7	41.5	0.91

^1^ Values are numbers (percentages) unless stated otherwise. IQR = interquartile range. Leisure-time physical activity (PA) levels are reported as means adjusted for age, sex, and energy intake. ^2^ Available for 21,889 participants. The *p*-values were obtained using generalized linear models both for continuous and categorical dependent variables adjusted for age, sex, and energy intake.

**Table 3 nutrients-15-03976-t003:** Risk of cancer hospitalization in association with pro-vegetarian food patterns (FP) in cancer-free participants from the Moli-sani study cohort (n = 22,081).

	Quintiles of Dietary Scores						
	Q1	Q2	Q3	Q4	Q5	P for Trend	1 SD Increment
**General pro-vegetarian FP**							
No. of cancer hospitalizations/no. of subjects	470/4720	407/4042	485/4670	448/4037	496/4612	-	-
Person-years, n	58,225	49,523	57,285	49,385	57,472	-	-
Event rates per 10,000 person-years	80.7	82.2	84.7	90.7	86.3	-	-
Model 1 (HR; 95%CI)	-1-	0.90 (0.79–1.03)	0.89 (0.78–1.01)	0.92 (0.80–1.04)	0.84 (0.74–0.95)	0.020	0.94 (0.90–0.98)
Model 2 (HR; 95%CI)	-1-	0.91 (0.80–1.04)	0.89 (0.78–1.01)	0.93 (0.82–1.06)	0.85 (0.75–0.97)	0.045	0.95 (0.91–0.99)
**Healthful pro-vegetarian FP**							
No. of cancer hospitalizations/no. of subjects	382/4017	512/5121	465/4263	501/4690	446/3990	–	–
Person-years, n	48,743	62,380	52,232	58,222	50,313	–	–
Event rates per 10,000 person-years	78.4	82.1	89.0	86.0	88.6	–	–
Model 1 (HR; 95%CI)	-1-	0.93 (0.82–1.07)	1.01 (0.88–1.15)	0.94 (0.83–1.08)	0.98 (0.85–1.12)	0.90	0.99 (0.95–1.03)
Model 2 (HR; 95%CI)	-1-	0.92 (0.80–1.05)	0.98 (0.85–1.12)	0.91 (0.80–1.05)	0.93 (0.81–1.07)	0.41	0.97 (0.93–1.01)
**Unhealthful pro-vegetarian FP**							
No. of cancer hospitalizations/no. of subjects	474/4427	509/4639	396/3901	490/4752	437/4362	–	–
Person-years, n	55,347	57,138	47,920	58,328	53,157	–	–
Event rates per 10,000 person-years	85.6	89.1	82.6	84.0	82.2	–	–
Model 1 (HR; 95%CI)	-1-	1.04 (0.91–1.17)	0.97 (0.84–1.10)	0.97 (0.86–1.10)	0.97 (0.85–1.11)	0.41	0.99 (0.95–1.03)
Model 2 (HR; 95%CI)	-1-	1.05 (0.93–1.19)	1.00 (0.87–1.14)	1.01 (0.89–1.15)	1.04 (0.91–1.18)	0.85	1.01 (0.97–1.06)

Hazard ratios (HR) with 95% confidence interval (95%CI) obtained from multivariable cause-specific Cox proportional hazards regression models. The multivariable-adjusted Model 1 was controlled for age, sex, energy intake, and alcohol intake (g/d; continuous). The multivariable-adjusted Model 2 was controlled for sex, age, energy intake, alcohol intake (g/d; continuous), residence, educational level, housing tenure, occupational class, smoking status, body mass index (categorical), leisure-time physical activity, history of CVD, diabetes, hypertension, hyperlipidemia, aspirin use, oral contraception use, hormone replacement therapy, menopausal status, and family history of cancer.

**Table 4 nutrients-15-03976-t004:** Subgroup and sensitivity analyses for the association of pro-vegetarian food patterns (FP) (1 SD increase) with risk of cancer hospitalization among cancer-free participants from the Moli-sani study cohort (n = 22,081).

		General Pro-Vegetarian FP			Healthful Pro-Vegetarian FP			Unhealthful Pro-Vegetarian FP		
	No. of Cases/ No. of Subjects	HR (95%CI)	*p*-Value	P for Interaction	HR (95%CI)	*p*-Value	P for Interaction	HR (95%CI)	*p*-Value	P for Interaction
*Subgroup analyses*										
Sex										
Men	1335/10,600	0.93 (0.88–0.98)	0.011	0.55	0.96 (0.90–1.01)	0.13	0.56	1.04 (0.98–1.10)	0.19	0.32
Women	971/11,481	0.97 (0.91–1.04)	0.45		0.98 (0.92–1.05)	0.62		1.00 (0.94–1.07)	0.92	
Age										
<65 y	1434/17,239	0.93 (0.88–0.98)	0.011	0.86	0.95 (0.90–1.00)	0.054	0.73	1.03 (0.97–1.08)	0.32	0.96
≥65 y	872/4842	0.94 (0.88–1.01)	0.10		0.97 (0.91–1.05)	0.49		1.02 (0.95–1.10)	0.57	
Body mass index										
Normal weight (≤25 kg/m^2^)	540/6119	0.93 (0.85–1.02)	0.11	0.83	0.93 (0.86–1.02)	0.11	0.59	1.10 (1.01–1.20)	0.035	0.12
Overweight (>25 ≤30 kg/m^2^)	991/9473	0.95 (0.89–1.02)	0.14		0.98 (0.91–1.04)	0.47		1.01 (0.95–1.08)	0.68	
Obese (>30 kg/m^2^)	775/6489	0.96 (0.89–1.04)	0.31		0.99 (0.92–1.07)	0.89		0.96 (0.90–1.04)	0.33	
Smoking status										
Non-smokers	919/10,950	0.96 (0.90–1.03)	0.27	0.94	0.99 (0.92–1.06)	0.72	0.80	1.00 (0.94–1.07)	0.89	0.28
Current smokers	588/5140	0.93 (0.85–1.01)	0.076		0.97 (0.89–1.05)	0.42		0.98 (0.90–1.07)	0.69	
Former smokers	799/5991	0.95 (0.89–1.03)	0.22		0.95 (0.88–1.02)	0.18		1.06 (0.99–1.14)	0.10	
*Sensitivity analyses*										
Excluding CVD	2143/20,943	0.95 (0.91–0.99)	0.019	−	0.97 (0.93–1.02)	0.21	−	1.00 (0.96–1.05)	0.84	−
Excluding diabetes	2122/21,010	0.94 (0.90–0.98)	0.0046	−	0.96 (0.92–1.01)	0.11	−	1.01 (0.97–1.06)	0.52	−
Excluding early cancer hospitalizations	2031/21,806	0.96 (0.91–1.00)	0.060	−	0.99 (0.94–1.03)	0.59	−	1.00 (0.96–1.05)	0.85	−

Hazard ratios (HR) with 95% confidence interval (95%CI) obtained from multivariable cause-specific Cox proportional hazards regression models including age, sex, energy intake, alcohol intake, residence, educational level, housing tenure, occupational class, smoking status, body mass index (categorical), leisure-time physical activity, history of CVD, diabetes, hypertension, hyperlipidemia, aspirin use, oral contraception use, hormone replacement therapy, menopausal status, and family history of cancer.

**Table 5 nutrients-15-03976-t005:** Pro-vegetarian food patterns (FP) and risk of hospitalization by cancer site in cancer-free participants from the Moli-sani study cohort (n = 22,081).

	General Pro-Vegetarian FP		Healthful Pro-Vegetarian FP		Unhealthful Pro-Vegetarian FP	
Cancer site	1 SD Increase HR (95%CI)	Q5 vs. Q1 HR (95%CI)	1 SD Increase HR (95%CI)	Q5 vs. Q1 HR (95%CI)	1 SD Increase HR (95%CI)	Q5 vs. Q1 HR (95%CI)
Respiratory tract (n = 183)	0.88 (0.76–1.03)	0.61 (0.38–0.98)	0.87 (0.75–1.02)	0.70 (0.41–1.19)	1.14 (0.98–1.33)	1.68 (1.06–2.68)
Digestive (n = 598)	0.88 (0.81–0.96)	0.74 (0.57–0.95)	0.92 (0.85–1.00)	0.76 (0.58–0.99)	1.00 (0.92–1.08)	0.92 (0.70–1.21)
Genitourinary organs (n = 395)	0.97 (0.87–1.08)	0.86 (0.63–1.18)	1.00 (0.90–1.11)	1.07 (0.76–1.51)	0.99 (0.90–1.10)	0.94 (0.68–1.29)
Breast (n = 285)	0.98 (0.87–1.10)	0.94 (0.66–1.36)	0.98 (0.87–1.10)	0.85 (0.58–1.24)	1.09 (0.96–1.22)	1.40 (0.95–2.05)
Prostate (n = 219)	1.08 (0.94–1.24)	1.37 (0.88–2.13)	1.07 (0.94–1.23)	1.37 (0.84–2.23)	1.03 (0.90–1.19)	1.21 (0.80–1.82)
Lymphatic and Hematopoietic Tissue (n = 178)	1.03 (0.88–1.20)	1.03 (0.65–1.63)	1.06 (0.90–1.23)	1.03 (0.631.71)	1.04 (0.90–1.22)	1.24 (0.77–2.00)
Brain and nervous system (n = 41)	0.93 (0.68–1.27)	0.92 (0.37–2.29)	1.15 (0.85–1.56)	1.36 (0.54–3.44)	1.04 (0.76–1.43)	0.82 (0.31–2.18)
Other (n = 407)	0.94 (0.86–1.05)	0.80 (0.59–1.08)	0.96 (0.87–1.06)	1.03 (0.73–1.44)	0.94 (0.85–1.04)	0.78 (0.56–1.08)

Hazard ratios (HR) with 95% confidence interval (95%CI) obtained from multivariable cause-specific Cox proportional hazards regression models including age, sex, energy intake, alcohol intake, residence, educational level, housing tenure, occupational class, smoking status, body mass index (categorical), leisure-time physical activity, history of CVD, diabetes, hypertension, hyperlipidemia, aspirin use, oral contraception use, hormone replacement therapy, menopausal status, and family history of cancer.

## Data Availability

The data underlying this article will be shared on reasonable request to the corresponding author. The data are stored in an institutional repository (https://repository.neuromed.it), and their access is restricted by ethical approvals and the legislation of the European Union.

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
