# Peer review of "Pro-Vegetarian Food Patterns and Cancer Risk among Italians from the Moli-Sani Study Cohort"

_nutrients, 2023, doi:10.3390/nu15183976_

Round 1

Reviewer 1 Report

This is important work to present in the journal, as reduction of cancer incidence is a public health priority worldwide. I understand the choice to consider potatoes as unhealthful (due to increased blood sugar response, unhealthful preparation methods, and prior research categorization of potatoes as unhealthful.) Still, I wish there were a way to determine whether the potatoes themselves or the method of preparation is truly the issue. While it is true that this work may be limited to an Italian population, it is a good step toward analysis in the larger Mediterranean region where individuals are more likely to adhere to the Mediterranean and/or pro-vegetarian diet culturally. 

In the abstract, first sentence could read "there is a paucity" to follow conventions of English where I live, but that may not be a true grammar error.

Page 2, third full paragraph: I think it would be more efficient to say, "the proportion of true vegetarians" or "the proportion of vegans" (if your intent is to say that not many people exclude all animal products from their diets.)

Page 3, second line should read, "disturbances in mental health or decision making impairments"

Page 4, first paragraph below the table, should "a priori" be italicized as in other places in the text?

Page 6, discussion of sensitivity analysis could be made parallel by stating, "test the robustness of the findings by excluding a) participants with a history of CVD, b) participants with diabetes, and c) participants whose cancer hospitalizations occurred within the first two years of follow-up."

Page 6, I believe you intend the abbreviation PVG to mean pro-vegetarian; you could include the abbreviation after the first mention of the term ("pro-vegetarian (PVG)") and then use the abbreviation the rest of the way through the paper.

Author Response

Thank you very much for your comments

Reviewer 2 Report

Authors present an interesting work as they operated on a very large sample, and over a long period, there are various points to improve:

Title: should be modified as, even from the data it does not emerge clearly whether it is the increase in vegetable consumption or the decrease in animal products that is effective, therefore a title with a diet with a higher intake of non-pro-vegetarian vegetables

Introduction: already from this point, a possible mechanism for which the outcome will be valid should be considered

Methods: the scoring of animal products is questionable; highly processed products such as mortadella or frankfurters and meat in general cannot be put in the same group; the same goes for fish.

Nothing is said about physical activity, a fundamental point

Results: there are very wide intervals, and this underlies a strong individual variability

Discussions: the patterns should be thoroughly analyzed; for example, a possible explanation of the efficacy of MD is the contribution of polyphenols which is greater if more vegetables are eaten and even if these are not processed.

Should a mechanism be proposed to confirm what has been highlighted, inflammation? Ros? Epigenetics?

This sentence should be characterized as "a diet that contains high amounts of healthful plant products and minimal amounts of animal products should be encouraged" or how many? Of both? And not only that, a plausible mechanism to increase vegetables exists, but why decrease products of animal origin? If chosen appropriately, they can also be considered in quantity.

It needs revision

Author Response

Thank you very much for your comments.

Round 2

Reviewer 2 Report

The authors answered my comments, but few is changed on the manuscript; the main concerns remain, and they did not include all my suggestions in the weakness.

just a revision
